# Reproducible disease phenotyping at scale: Example of coronary artery disease in UK Biobank

Riyaz S. Patel[1,2]⊙*, Spiros Denaxas[ID][3,4]⊙*, Laurence J. Howe[1,2]⊙, Rosalind M. Eggo[4,5,6], Anoop D. Shah[2,3,4], Naomi E. Allen[7,8], John Danesh[9,10], Aroon Hingorani[1,2], Cathie Sudlow[11,12,13], Harry Hemingway[ID][2,3,4]

1 Institute of Cardiovascular Sciences, University College London, London, United Kingdom, 2 NIHR University College London Biomedical Research Centre, University College London and University College London Hospitals NHS Foundation Trust, London, United Kingdom, 3 Health Data Research UK London, University College London, London, United Kingdom, 4 Institute of Health Informatics, University College London, London, United Kingdom, 5 Faculty of Epidemiology and Public Health, London School of Hygiene & Tropical Medicine, London, United Kingdom, 6 Health Data Research UK London, London School of Hygiene and Tropical Medicine, London, United Kingdom, 7 Clinical Trial Service Unit and Epidemiological Studies Unit, Nuffield Department of Population Health, University of Oxford, Oxford, United Kingdom, 8 UK Biobank Ltd, Stockport, United Kingdom, 9 Health Data Research UK Cambridge, Hinxton, United Kingdom, 10 Department of Public Health and Primary Care, Cambridge University, Cambridge, United Kingdom, 11 Health Data Research UK Scotland, Edinburgh, United Kingdom, 12 Centre for Clinical Brain Sciences, University of Edinburgh, Edinburgh, United Kingdom, 13 HDR UK London, London School of Hygiene and Tropical Medicine, London, United Kingdom

⊙ These authors contributed equally to this work.
* Riyaz.Patel@ucl.ac.uk (RSP); s.denaxas@ucl.ac.uk (SD)

**Data Availability Statement:** Data cannot be shared publicly by the authors because of information governance restrictions around health data. The data can however be downloaded

## Abstract

### Importance

A lack of internationally agreed standards for combining available data sources at scale risks inconsistent disease phenotyping limiting research reproducibility.

### Objective

To develop and then evaluate if a rules-based algorithm can identify coronary artery disease (CAD) sub-phenotypes using electronic health records (EHR) and questionnaire data from UK Biobank (UKB).

### Design

Case-control and cohort study.

### Setting

Prospective cohort study of 502K individuals aged 40–69 years recruited between 2006–2010 into the UK Biobank with linked hospitalization and mortality data and genotyping.

### Participants

We included all individuals for phenotyping into 6 predefined CAD phenotypes using hospital admission and procedure codes, mortality records and baseline survey data. Of these,

following a project approval process by the UK Biobank. Researchers wishing to access the data can apply directly to the UK Biobank https://www.ukbiobank.ac.uk/enable-your-research/apply-for-access and the process involves registering on the access management system, submitting a research study protocol and paying a fee directly to the UK Biobank. The authors of this study did not receive any special privileges in accessing the data. UK Biobank is an open access research resource for researchers and accepts applications with no restriction.

**Funding:** This study is funded by a British Heart Foundation Intermediate Fellowship awarded to RSP (FS/14/76/30933). This research was also supported by the National Institute for Health Research University College London Hospitals Biomedical Research Centre through a grant awarded to SD and HH (UCLHBRC). Work at the University College London Institute of Health Informatics and Institute of Cardiovascular Science is also supported by a British Heart Foundation Accelerator Award given to SD (AA/18/6/24223). Health Data Research UK also supported the study through a grant awarded to SD, HH, CS, and JD (LOND1).

**Competing interests:** The authors have declared that no competing interests exist.

**Abbreviations:** MI, myocardial infarction; CAD, coronary Artery Disease; EHR, Electronic Health Records; UKB, UK Biobank; UA, Unstable Angina; CABG, Coronary Artery Bypass Grafting; PCI, Percutaneous Coronary Intervention.

408,470 unrelated individuals of European descent had a polygenic risk score (PRS) for CAD estimated.

## Exposure

CAD Phenotypes.

## Main outcomes and measures

Association with baseline risk factors, mortality (n = 14,419 over 7.8 years median f/u), and a PRS for CAD.

## Results

The algorithm classified individuals with CAD into prevalent MI (n = 4,900); incident MI (n = 4,621), prevalent CAD without MI (n = 10,910), incident CAD without MI (n = 8,668), prevalent self-reported MI (n = 2,754); prevalent self-reported CAD without MI (n = 5,623), yielding 37,476 individuals with any type of CAD. Risk factors were similar across the six CAD phenotypes, except for fewer men in the self-reported CAD without MI group (46.7% v 70.1% for the overall group). In age- and sex- adjusted survival analyses, mortality was highest following incident MI (HR 6.66, 95% CI 6.07–7.31) and lowest for prevalent self-reported CAD without MI at baseline (HR 1.31, 95% CI 1.15–1.50) compared to disease-free controls. There were similar graded associations across the six phenotypes per SD increase in PRS, with the strongest association for prevalent MI (OR 1.50, 95% CI 1.46–1.55) and the weakest for prevalent self-reported CAD without MI (OR 1.08, 95% CI 1.05–1.12). The algorithm is available in the open phenotype HDR UK phenotype library (https://portal.caliberresearch.org/).

## Conclusions

An algorithmic, EHR-based approach distinguished six phenotypes of CAD with distinct survival and PRS associations, supporting adoption of open approaches to help standardize CAD phenotyping and its wider potential value for reproducible research in other conditions.

## Introduction

The creation and maturation of very large biobanks including the UK Biobank (UKB) [1]. China Kadoorie Biobank [2], US Million Veterans Program [3], All of Us Research Program [4] and many others – offers unique opportunities to better understand genetic, lifestyle and environmental factors that underpin development of common complex conditions, such as coronary artery disease (CAD). However, consistently, and accurately ascertaining participants who have prior or incident diseases presents major challenges. This is because data reported by participants at baseline may be of variable quality and reliability. Furthermore, in population biobanks involving several hundreds of thousands of participants, it is not practicable to perform individual case-based adjudication at scale, as is customary in traditional clinical trials [5].

One solution is to use electronic health records (EHR), with International Classification of Diseases (ICD) coded data on diagnoses, hospitalisations, procedures and deaths, alone or in combination with selected baseline survey data to enhance phenotyping of disease cases and controls [6]. While this approach is widely used, a lack of internationally agreed standards for selecting and combining ICD coded data for defining common diseases and their clinically

relevant sub-types has resulted in significant variation in practice with often multiple different ways of combining ICD codes for defining the same disease [7]. Furthermore, aggregating all available disease entries, to increase sample size, could potentially induce heterogeneity in association estimates as different data sources may have different case misclassification rates, and thus paradoxically risk loss of statistical power.

Importantly, recent attention on the "reproducibility crisis" in medicine highlight the importance of open and accessible definitions of disease [8]. The use of large scale EHR data in particular has come under particular scrutiny in this regard within the last year [8,9]. UKB offers opportunities to develop and evaluate openly standardised EHR phenotyping algorithms [1]. First, as it is widely-accessible, it provides an excellent opportunity to inform international research practice, with a community of >15,000 research users globally. Second, it combines exceptional scale and detail, offering an opportunity to evaluate the performance of such algorithms through association. Manual adjudication of cases is not feasible in many large scale studies, and previous studies have shown the importance of using information on prognosis (mortality) and genetic association to provide evidence of the extent to which EHR phenotypes reproduce findings from research using adjudicated case definitions [10]. Third, UKB data on self-reported illnesses collected via the baseline assessment provides an opportunity to evaluate the extent to which self-reported data may add to or dilute case definitions.

Using CAD as an exemplar high profile condition, and extending prior work by UKB [11–14], we sought to develop, evaluate and share EHR phenotype algorithms for CAD, distinguishing clinically relevant sub-types. Such an approach to define CAD has thus far not been reported yet could have important research and even clinical utility. We focus primarily on ICD coded hospital admission and mortality data supplemented with self-reported survey data from the UKB. We define six distinct phenotypes through an algorithmic approach: EHR defined Incident myocardial infarction (MI); prevalent MI; incident CAD without (MI; prevalent CAD without MI; prevalent self-reported MI; prevalent self-reported CAD without MI. We then sought to assess the fidelity of these CAD phenotypes, by evaluating their associations with known risk factors, long term mortality and a polygenic risk score for CAD.

To facilitate wider adoption and encourage standardization of CAD phenotyping, we make the algorithm and its outputs openly available for researchers to review and use in UK Biobank and other large-scale cohorts (available at the Health Data Research UK phenotype library (https://portal.caliberresearch.org/).

## Methods

### Data sources & linkage

**UK Biobank baseline recruitment.** UKB is a prospective cohort study of 502K individuals aged 40–69 years recruited between 2006–2010 across England, Wales and Scotland [1]. Study participants completed a questionnaire and a nurse-led interview at baseline and were also followed-up for health outcomes via linked EHR [15,16]. All participants provided written informed consent and ethics approval was granted from the North West Multi-Centre Research Ethics Committee (06/MRE08/65).

**Hospital admission data.** Hospital admission data in UKB is obtained from the Hospital Episode Statistics (HES) database, the Patient Episode Database for Wales (PEDW) database and the Scottish Morbidity Record-01 (SMR-01) in England, Wales, and Scotland, respectively. International Statistical Classification of Diseases and Health-Related Problems, 10th revision (ICD-10) classifications are used to record primary and secondary diagnoses (note that ICD-10 classifications in the UK are distinct from ICD-10 CM) [6]. Office of Population Censuses and Surveys Classification of Interventions and Procedures (OPCS) Version 4 classifications

are used to record procedures, similar to the Current Procedural Terminology (CPT) advocated by the American Medical Association [17].

**Mortality follow up.** UKB obtains linked data on mortality by cause (underlying and contributing causes) from national death registries, via the Office of National Statistics data (ONS) for England and Wales and the General Register Office database (GRO) for Scotland.

## Coronary artery disease phenotypes

A cardiologist (RP), informatician (SD) and epidemiologist (RME), led the development of a rules-based algorithm to identify individuals, sequentially, through available linked EHR data sources and baseline UKB data, based on their recorded diagnoses or procedures relating to the following CAD phenotypes:

1. All CAD (any CAD/MI relevant code or self reported diagnosis at any time). Sub-phenotypes include:

    i. Incident (new onset) MI—identified through EHR and defined as an MI occurring after enrolment into UKB

    ii. Prevalent (pre-existing) MI—identified through EHR and defined as an MI occurring prior to enrolment into UKB

    iii. Prevalent self-report MI—identified at nurse-led interview and defined as a self reported MI occurring prior to enrolment into UKB

    iv. Incident CAD without MI—identified by EHR and defined as a CAD occurrence after enrolment into UKB

    v. Prevalent CAD without MI—identified by EHR or self reported procedures, defined as occurring before enrolment into UKB

    vi. Prevalent self-report CAD without MI—identified at nurse-led interview, defined as a CAD occurrence before enrolment into UKB

2. Controls were free from any CAD at any time

When both a prevalent and an incident diagnosis exist, the option is available to classify as either depending on the research question and analysis plan. For the purposes of this analysis, we have classified those individuals by their more recent (incident) events when there is greater certainty of date and diagnosis.

## Diagnosis codes

All CAD phenotypes were defined using ICD10, OPCS and self-reported diagnoses, medications, and procedures, by a combination of expert consensus from practicing UK clinicians and review of existing literature. Specific codes and criteria to define each phenotype are listed in **S1**–**S3 Tables**. Phenotypes were defined using knowledge of the databases, their coverage and quality. Since not all CAD events or diagnoses are captured in any single source, we planned *a priori* to develop strategies to maximize the yield of cases using a hierarchical approach, whereby cases not found in one database could be searched for in subsequent sources.

## Algorithm structure

Details of the algorithm are provided in Supplementary methods. Briefly, in a hierarchical order, individuals with a coded diagnosis of MI in EHR from secondary care or mortality

records were first identified and categorised (see supplementary methods for code lists) (**Fig 1**). If no EHR coded diagnoses of MI was made, individuals were then classified as CAD without MI if they matched to code lists identifying presence of CAD, such as revascularization procedures (see supplement). Further classification based on date of enrolment categorized individuals as having prevalent (diagnoses prior to date of enrolment) or incident (occurring after enrolment) disease. If any MI or CAD without MI code was identified, the individual was censored from further classification.

If individuals did not have one of the MI or CAD EHR codes, or a self-reported procedure (see supplementary methods), then a self-report diagnosis for "heart attack" or "angina" was used to classify remaining individuals as self-report MI and self-report CAD without MI, respectively. Finally, those who did not appear in any EHR source or self-report CAD were labelled as "never CAD" (if they since died) or "no CAD" (if they were still alive); and both groups were used as controls free from CAD **Fig 1.**

All linked hospital admissions and mortality datasets were censored on November 1st, 2016, to maintain alignment and ensure completeness of follow-up across the available data sources. On publication, algorithm-generated phenotypes will be returned to UK Biobank to be included in the data showcase for use by the research community and will in parallel be included in the HDR UK Phenotyping platform.

## Statistical analyses

**Risk factor & characteristics.**   We reported baseline characteristics for participants categorised into each of the CAD phenotypes, including the All-CAD group as well as controls. We present proportions for categorical variables and means with standard deviations for continuous traits.

**Mortality risk.**   We compared survival between algorithm classes from the date of UK Biobank recruitment for prevalent cases and controls and from hospital admission date for incident cases. We evaluated all-cause mortality risk for the six algorithmically generated CAD phenotypes and the aggregated All CAD phenotype using Kaplan- Meier survival analysis and Cox proportional- hazards models. All models were adjusted for age and sex and Schoenfeld residuals were checked to ensure non-violation of the proportional- hazards assumption.

**Polygenic risk score.**   A CAD PRS was derived from a previous genome-wide association study (GWAS) of CAD in European populations, independent of UK Biobank, which compared (mostly prevalent) cases (MI, acute coronary syndrome, chronic stable angina or coronary stenosis > 50%) to controls [18]. The PRS of 182 SNPs was constructed in a sample of 408,470 unrelated individuals of European descent using independent SNPs associated with CAD (P<$5 \times 10^{-6}$) after LD clumping (parameters: $r^2$ = 0.2, 250kb) in PLINK v1.9 [19]. Logistic regression was used to evaluate associations between the six algorithmically generated CAD phenotypes and the aggregated all CAD phenotype against the CAD free controls, with age and sex included as covariates.

All analyses were conducted using R statistical software [20].

## Results

### Overlap between data sources

We first examined the overlap between baseline UKB survey data and EHR coded diagnoses for prevalent MI and CAD diagnoses (n = 36,015). Of the 12,006 individuals self-reporting MI at baseline, only 2,341 had a relevant MI or CAD ICD code in hospital EHR data at study enrolment (prevalent MI). Similarly of the 16,789 individuals self-reporting CAD at baseline (of which 5,048 also self-reported MI), only 3,307 could be identified with a MI or CAD EHR coded entries. Of

note, 9752 individuals were identified as having either prior MI or CAD at baseline, through EHR coded diagnoses but had not reported either in the baseline questionnaire. **Fig 2**.

## Algorithm implementation

Starting with the entire UKB population (n = 502,631), the algorithm shown in **Fig 1** sequentially classified all individuals into the six sub-phenotypes: Prevalent MI (n = 4,900); incident

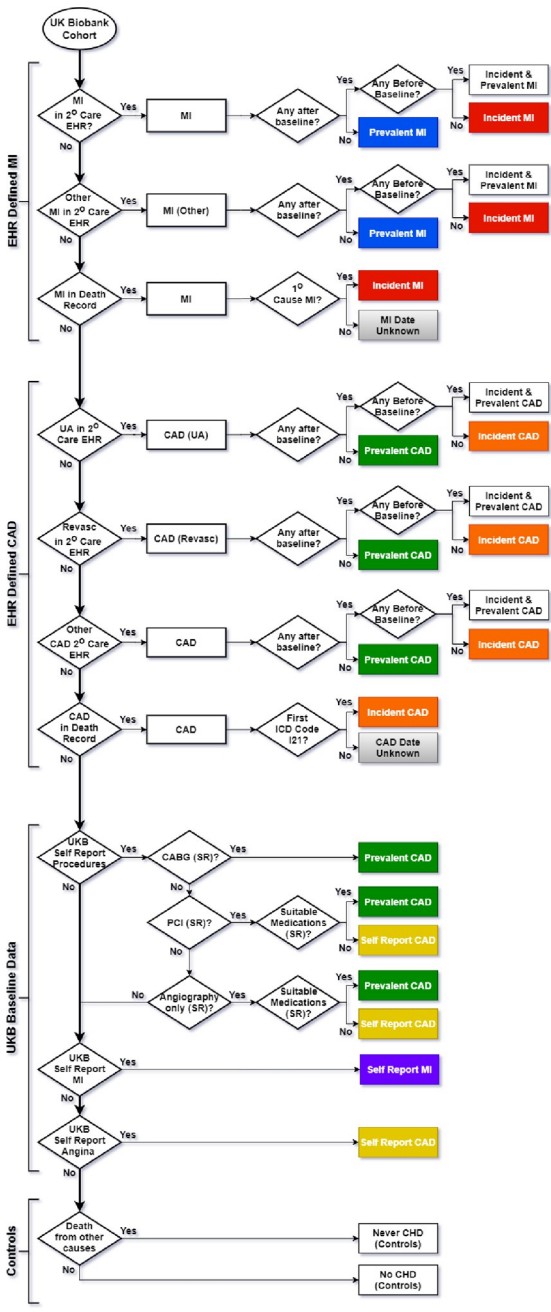

**Fig 1. Phenotyping algorithm to generate an All-CAD phenotype and CAD free controls along with 6 sub-phenotypes of CAD.** The algorithm is designed to be run either as a complete run as presented, or as modules taking each of the sections and running individually. Where incident and prevalent MI or CAD is present a choice can be made to designate either as prevalent or incident. ICD and related Codes used within each section are provided in Supplementary Tables.

MI (n = 4,621), prevalent CAD without MI (n = 10,910), incident CAD without MI
(n = 8,668), self-reported MI (n = 2,754); self-reported CAD without MI (n = 5,623). Combining all of these into an aggregated all CAD phenotype, including prevalent and incident cases, yielded a total of 37,476 individuals with CAD and 465,155 controls free of any CAD.

### Risk factors and characteristics

Risk factors and characteristics for each sub phenotype, including the aggregated all CAD phenotype and CAD free controls are presented in S4 Table. Broadly, compared to controls, those with any of the CAD phenotypes, were more likely to be male, older, taking a statin and had higher levels of known CAD risk factors, including smoking prevalence, diabetes, weight, and socio-economic deprivation. The only exception was for mean systolic BP which was marginally lower among those with prevalent MI compared to controls.

Within the CAD groupings, mean age was similar across all six phenotypes but there were substantially fewer men in the self-report CAD without MI group compared to the All-CAD group (46.7% v 70.1%). Prevalence of other risk factors were broadly similar, although there was a trend to greater smoking in the MI compared with the CAD without MI phenotypes (31–36% v 21.1%-25.4%).

### Mortality associations

Amongst 502,631 UK Biobank participants, median follow up to death or 1 November 2016 was 7.8 years (IQR 1.04) from study recruitment date, and a total of 14,419 deaths occurred during follow-up. Of these 3,770 deaths (10.1%) were among those with any CAD diagnosis (from baseline for prevalent cases and time of first event for incident cases) and 10,649 (2.3%) occurred among controls without any CAD diagnoses.

Kaplan Meier analysis for the CAD sub-phenotypes revealed a gradient in risk of mortality across the six CAD sub-phenotypes (Fig 2). The highest risk was observed for incident MI cases during the first year following the incident MI, with a lower but still substantial risk for incident CAD without MI as shown in Fig 2A. For both prevalent and self-reported phenotypes, with follow-up from the date of recruitment, the risks were substantially lower than for incident events, and over a longer follow up period (Fig 2B).

We found a clear gradient in risk for the sub-phenotypes in age and sex adjusted association estimates for mortality when compared to CAD free controls (Fig 3). We found the highest risk in incident MI cases (HR 6.66; 95% C.I. 6.07 and 7.31), followed by incident CAD without MI (HR 5.65; 95% C.I. 5.28, 6.04), prevalent MI (HR 2.52; 95% C.I. 2.29, 2.78) and prevalent CAD without MI (HR 1.75; 1.62, 1.89). Self-reported only prevalent MI cases (HR 1.77; 95% C.I. 1.53, 2.06) had a similar relative risk to EHR defined prevalent CAD without MI, while self-reported only prevalent CAD without MI cases had the lowest relative mortality risk (HR 1.31; 95% C.I. 1.15,1.50) of all sub-phenotypes when compared to controls. The aggregated all CAD phenotype, incorporating all sub-phenotypes, had an intermediate relative risk (HR 2.74; 95% CI 2.63–2.86) for mortality (Fig 3).

### Polygenic risk score association

Polygenic risk score associations with CAD have been widely reported using data from UKB participants of European ancestry. We generated a CAD PRS for the same population (n = 408,470) to evaluate its association with our generated phenotypes. We found it was robustly associated with the aggregated all CAD phenotype when compared to controls, consistent with prior reports (OR 1.34; 95% CI 1.32–1.35) after adjustment for age and sex (Fig 4). However, among the six sub-phenotypes there was a clear graded association, with the largest

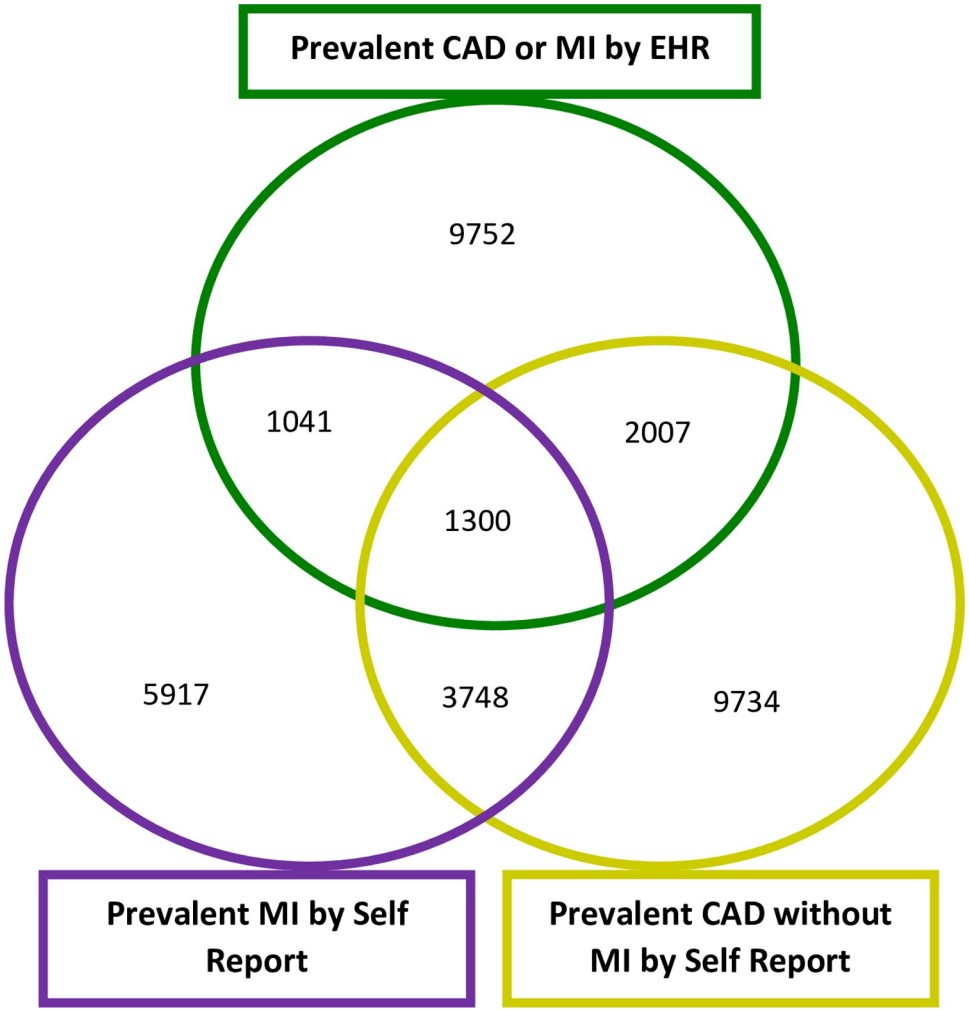

**Fig 2. Overlap of self-reported diagnoses in UKB and those identified through EHR, for prevalent CAD or MI.**
Venn diagram showing the overlap between UKB survey derived self-reported MI cases, self-reported CAD without
MI cases and CAD or MI cases identified in EHR.

effect size noted found for association with the algorithmically generated prevalent MI (OR
per SD increase in PRS 1.50; 95% CI 1.46–1.55) and the prevalent CAD without MI (OR per
SD increase in PRS 1.44; 95% CI1.41–1.47) phenotypes. The weakest association was for self-
report CAD without MI (OR per SD increase in PRS 1.08; 95% CI 1.05–1.12) (**Fig 5**).

## Discussion

In response to the challenge of reproducibly phenotyping diseases at scale using multimodal
data, we have developed, evaluated, and shared a reproducible method of identifying individu-
als with CAD and its subtypes, as an exemplar, in a large national biobank, using a combina-
tion of ICD-coded hospitalization diagnoses and procedures, mortality and baseline
participant self-reported data. We evaluated the fidelity of these phenotypes by demonstrating
distinct and expected associations with mortality and a PRS for CAD. The clear gradient in
association estimates for these sub-phenotypes further illustrates an important limitation of
aggregating all CAD data into a single phenotype. We have made the algorithm output and

**A) Incident Cases**

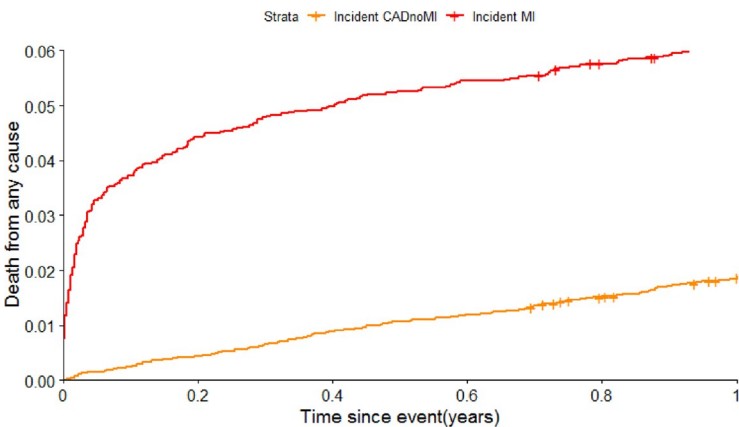

**B) Prevalent Cases**

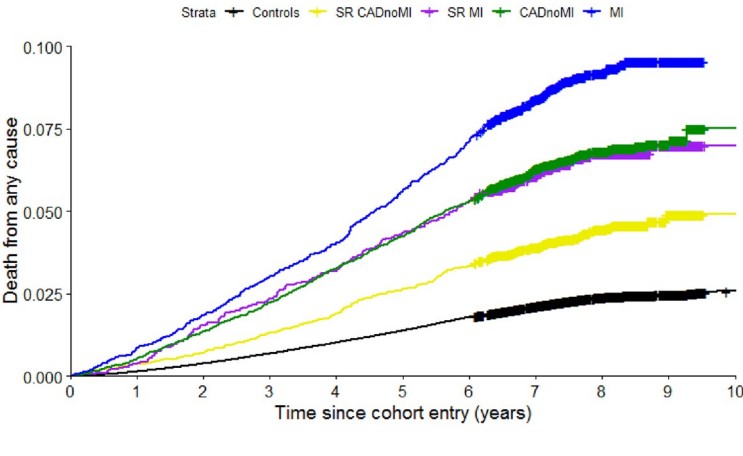

**Fig 3. Kaplan Meier survival analysis for of CAD phenotypes with all-cause mortality. Panel A** shows survival curves for participants identified to have an incident CAD and MI phenotype during follow up, with baseline recorded as the date of the clinical event. **Panel B** shows survival curves for participants identified as having prevalent CAD and MI phenotypes, with baseline recorded as date of enrolment.

code lists publicly available, to encourage more precise and reproducible sub-phenotyping of CAD and to promote the wider adoption of this phenotyping approach for other conditions where similar data are available in cohorts nationally and internationally.

Algorithmic validation in the classical sense would include the expert, manual review of case notes and results from ECGs and the calculation of positive predictive value (PPV) and negative predictive value estimates (NPV). These data however are not available in large-scale population studies and EHR data such as the UK Biobank nor does the approach scale with large numbers of participants. We have therefore followed a robust evaluation approach which has been previously applied and validated in large scale EHR data that span care settings [10]

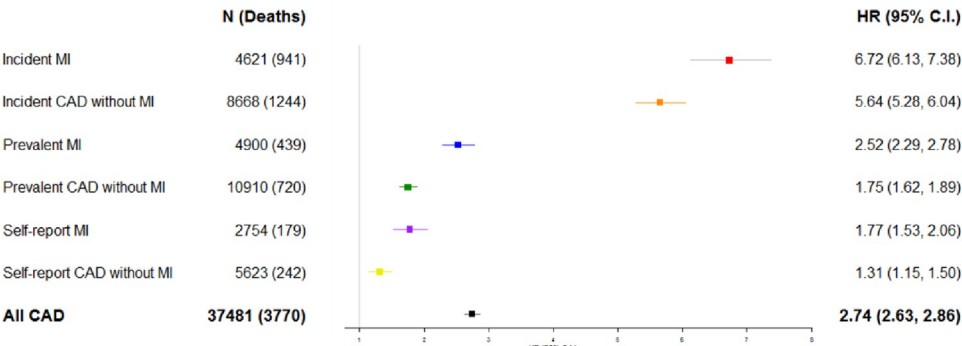

**Fig 4. Coronary artery disease phenotypes and mortality.** Hazard ratios (HR) presented for all-cause mortality (95% C.I.) for CAD phenotypes; Adjusted for age and sex, compared to 465,155 CAD free controls (never or no CAD).

which provides multiple layers of evaluation through the replication of existing epidemiological findings and, in this case, a PRS for CAD. These multiple layers of complementary information provide further evidence towards the validity and robustness of the phenotyping algorithms presented here.

We and others have previously demonstrated the value of using linked EHR data to improve identification of both prevalent and incident diseases at scale [10,21]. Building on early work in the UK Biobank for identifying CAD [11,12], we have shown that an expanded rules-based algorithm integrating multiple data sources, with clinical and domain experts guiding code selection and a team familiar with both the specialty and coding practices in the UK, can identify multiple CAD sub-phenotypes in the UKB. While further work is needed to incorporate the added information available in primary care and other data sources, for now we advocate use of this algorithm for improving and standardising CAD phenotyping in UKB and other population-based cohorts using widely available data sources for ascertaining disease status. We also encourage investigators to consider selecting and working with selected sub-phenotypes of CAD rather than an aggregated all CAD phenotype, which may yield different biological and prognostic insights.

Our algorithm generates six pre-specified CAD sub-phenotypes as well as an aggregated outcome for all CAD phenotype and CAD free controls [22,23]. We confirmed that CAD cases identified through the algorithm were representative of a CAD population, being mostly male

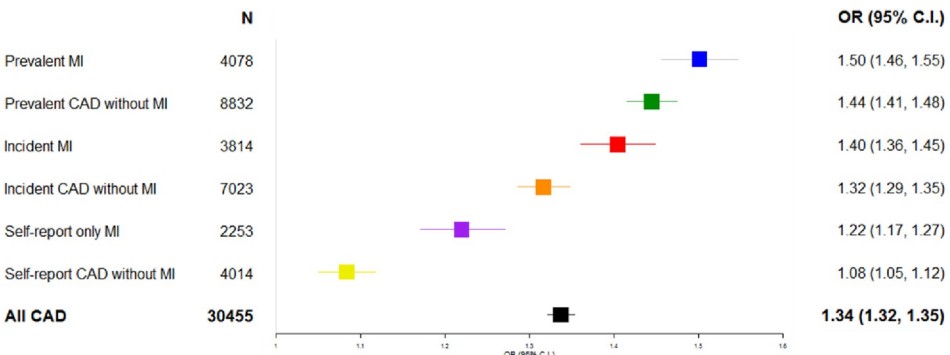

**Fig 5. Coronary artery disease phenotypes and association with polygenic risk score for CAD.** Odds Ratios (ORs) per 1 S.D. increase in CAD PRS, (95% C.I.) for CAD phenotypes; Adjusted for age and sex, compared to 378,025 CAD free controls (never or no CAD).

in their 60s, with higher prevalence of diabetes and smoking compared to controls. Across the six sub-phenotypes, there were some differences such as a lower mean systolic blood pressure in prevalent MI cases compared to the other categories, perhaps arising due to treatment. Furthermore participants who had only self-reported CAD without MI (i.e. without any verifying EHR or procedural data) were predominantly female and involved fewer smokers, a sub-population consistent with participants who have had chest pain or angina but without epicardial disease [24–26]. Overall, these findings support the robustness of the CAD phenotype and sub-phenotypes identified by our algorithms and highlight the potential for dilution of associations in investigations that include patients with self-reported CAD only.

Of note, we found a near three-fold increase in risk of death among those with any CAD (aggregated phenotype) compared to disease free controls. However, there was a steeper gradient in mortality associations with those having an algorithm-identified incident MI at greatest risk (6-fold) of near term death following an event, as has been demonstrated previously [27]. Those surviving a prior MI before entry into the cohort, were at relatively lower risk, which may be explained by selection bias, with preferential recruitment of stable survivors who have had a more remote history of MI, into UKB. Those only self-reporting an MI without EHR corroboration, had a higher risk than controls. This may be due to greater risk factor burden in these participants as we have shown above, or because in some cases, EHR may not have captured true self-reported events as they occurred prior to the start of the EHR systems (1997–1998 in England and Wales; 1981 Scotland), occurred abroad or were treated in the private sector and as such risk was higher in some participants. Nonetheless, those with self-report MI still had a lower risk than those with EHR corroboration and this may be due to erroneous participant understanding leading to false positive self-reporting [28,29].

Polygenic risk scores have been widely evaluated in the UKB consistently demonstrating association with a variably defined and aggregated single CAD phenotype. When we examined associations with a CAD polygenic risk score, we noted that for both algorithmically generated prevalent CAD and MI cases, there was a stronger association than with incident cases. These findings likely reflect the fact that the score is derived from genome wide significant variants in studies of predominantly prevalent cases of both CAD and MI (e.g. CARDIoGRAM-PlusC4D) [18,23,30]. We and others have also shown weaker associations for genetic variants with incident CAD compared to prevalent CAD [31,32]. Furthermore, those identified as self-report only cases by the algorithm showed a significantly attenuated association with the PRS indicating a potential for misclassification when included as CAD cases in such studies. Use of self-reported cases–either alone or combined with EHR data, as a means to increase power, may attenuate effect estimates and could–paradoxically–reduce statistical power despite increasing the number of cases making the discovery of smaller effect sizes difficult.

The challenge of using EHR for disease phenotyping in a reproducible manner by researchers across the globe expands to all common diseases and as such our work also has relevant value beyond CAD. Recent initiatives in partnership with HDR UK have highlighted the need to offer guidance on defining prevalent and incident disease. For example, the variation in phenotyping approaches to conditions such as asthma have been well documented with 66 different EHR based algorithms identified in the literature to define asthma cases [7]. We and others have developed resources that help systematically define phenotypes using EHR data, and encourage researchers and domain experts to build on these and help define international standards [10]. Importantly, we have also shown that robust evaluation of algorithm performance, without need for case- based review, is possible by comparing estimates of exposure, mortality and genetic risk association, with prior estimates from other records based cohorts or consortia. Such associations are applicable to many conditions such as asthma, COPD, schizophrenia, among others.

## Limitations

Our study has some limitations. First, our phenotyping algorithm lacks full coverage of all EHR data sources for the capturing of CAD, such as primary care data which has recently been made available in the UKB. However, by restricting our work to ICD and OPCS-4 coded hospital data we anticipate greater potential for wider application to other cohorts given the availability and use of this coding system internationally. Secondly, other features that would better identify MI cases such as ECGs or clinical biomarkers were not available in the national linked hospitalization records. Recent large scale EHR initiatives in the UK, offer some promise towards capturing these data in the future [33]. Finally, individual case note review "validation" in the classical sense was unavailable to confirm algorithm validity and reliability and estimate positive predictive values, as UKB is independent of the health system unlike for example the MVP in the USA [3]. While UKB has coordinated some expert clinician led validation studies involving direct review of the full EHR for several health-related outcomes, these studies are time consuming and difficult to scale [34,35]. Recent progress in accessing unstructured clinical data for research in the UK may offer an opportunity to do this at scale in the future [10].

## Clinical implications

With increasing use of EHR across healthcare systems, there is an expectation to use coded data to continuously improve health care delivery quality and outcomes, with models of learning health systems [36]. Within cardiology services, clinicians and hospital administrators often seek to identify CAD patients in their hospitals, to measure or audit performance against standard guidelines (e.g. statin use among those with known CAD) or for inclusion in disease registries. Recent guidelines have also delineated acute MI into subtypes, and more recently stable CAD has recently been renamed to chronic coronary syndromes (CCS), further supporting the clinical relevance of sub-phenotyping of CAD for prognostic and treatment implications [37,38]. Our modular algorithm is well suited for implementation in such clinical settings and is arguably even more urgently needed to avoid the ad-hoc and non-standardized approaches in use today.

## Conclusion

In conclusion, we have developed and demonstrated the feasibility of deploying an algorithmic approach for combining multimodal data for disease phenotyping in a large national biobank. Specifically, for our exemplar condition of CAD, we derived 6 sub-phenotypes and demonstrated their validity through prognostic and genetic association techniques. At the same time, we found a clear gradient in effect sizes illustrating a major risk in aggregating CAD phenotypes, with a paradoxical loss of statistical power and/or diminished effect estimates. We encourage investigators utilizing the UKB for CAD research to use the freely available code to phenotype and sub phenotype CAD for more robust and reproducible analyses. We anticipate the principles of our approach to be applicable to all large-scale biobanks with similarly available EHR data and to other common diseases.

## Supporting information

**S1 Table. List of ICD-10 and ICD-9 terms used to define the MI phenotype.**
(DOCX)

**S2 Table. List of EHR codes used to define the CAD phenotyp.**
(DOCX)

**S3 Table. UKB available codes for CAD and MI diagnoses, medications, and procedures.**
(DOCX)

**S4 Table. Baseline participant characteristics of coronary disease phenotype samples.**
(DOCX)

**S1 File.**
(DOCX)

## Author Contributions

**Conceptualization:** Riyaz S. Patel, Anoop D. Shah, Naomi E. Allen, John Danesh, Cathie Sudlow, Harry Hemingway.

**Data curation:** Spiros Denaxas, Laurence J. Howe, Rosalind M. Eggo, Anoop D. Shah.

**Formal analysis:** Laurence J. Howe, Rosalind M. Eggo.

**Funding acquisition:** Spiros Denaxas, Harry Hemingway.

**Supervision:** Spiros Denaxas, Naomi E. Allen, John Danesh, Aroon Hingorani, Cathie Sudlow, Harry Hemingway.

**Visualization:** Laurence J. Howe, Rosalind M. Eggo.

**Writing – original draft:** Laurence J. Howe, Rosalind M. Eggo, Anoop D. Shah.

**Writing – review & editing:** Spiros Denaxas, Laurence J. Howe, Rosalind M. Eggo, Anoop D. Shah, Naomi E. Allen, John Danesh, Aroon Hingorani, Cathie Sudlow, Harry Hemingway.

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
