## [Decision Letter · Decision Letter 0]

22 Dec 2021

PONE-D-21-39043Reproducible disease phenotyping at scale: example of coronary artery disease in UK BiobankPLOS ONE

Dear Dr. Denaxas,

Thank you for submitting your manuscript to PLOS ONE. After careful consideration, we feel that it has merit but does not fully meet PLOS ONE’s publication criteria as it currently stands. Therefore, we invite you to submit a revised version of the manuscript that addresses the points raised during the review process.

We look forward to receiving your revised manuscript.

Kind regards,

Dylan A Mordaunt, MB ChB, FRACP, FAIDH

Academic Editor

PLOS ONE

Journal Requirements:

When submitting your revision, we need you to address these additional requirements. 1. Please ensure that your manuscript meets PLOS ONE's style requirements, including those for file naming. The PLOS ONE style templates can be found at https://journals.plos.org/plosone/s/file?id=wjVg/PLOSOne_formatting_sample_main_body.pdf and https://journals.plos.org/plosone/s/file?id=ba62/PLOSOne_formatting_sample_title_authors_affiliations.pdf
 2. We note that the grant information you provided in the ‘Funding Information’ and ‘Financial Disclosure’ sections do not match.  When you resubmit, please ensure that you provide the correct grant numbers for the awards you received for your study in the ‘Funding Information’ section. 3. Please amend either the abstract on the online submission form (via Edit Submission) or the abstract in the manuscript so that they are identical. 4. Please ensure that you refer to Figure 5 in your text as, if accepted, production will need this reference to link the reader to the figure. 5. Please include captions for your Supporting Information files at the end of your manuscript, and update any in-text citations to match accordingly. Please see our Supporting Information guidelines for more information: http://journals.plos.org/plosone/s/supporting-information. 

Additional Editor Comments:

Thank you for your submission. With specific regards to the criteria for publication:

1. The study appears to present the results of original research.

2. Results do not appear to have been reported published elsewhere.

3. Experiments, statistics, and other analyses are performed to a high technical standard however suggestions are made by the reviewer below.

4. Conclusions are presented in an appropriate fashion and are supported by the data.

5. The article is presented in an intelligible fashion and is written in standard English.

6. The research meets all applicable standards for the ethics of experimentation and research integrity.

7. The article adheres to appropriate reporting guidelines and community standards for data availability.

Reviewers' comments:

Reviewer's Responses to Questions

**Comments to the Author**

1. Is the manuscript technically sound, and do the data support the conclusions?

Reviewer #1: No

2. Has the statistical analysis been performed appropriately and rigorously? 

Reviewer #1: No

3. Have the authors made all data underlying the findings in their manuscript fully available?

Reviewer #1: Yes

4. Is the manuscript presented in an intelligible fashion and written in standard English?

Reviewer #1: Yes

5. Review Comments to the Author

Reviewer #1: This paper seeks to provide an algorithm for the classification of incident and prevalent CAD primarily using EHR coded data, supported by other coded data sources. There are several issues that need to be addressed to improve the utility of this work.

There was no comparison of the algorithm's classification to a "gold standard" classification. Hence, there was no "validation" in the understanding of most readers of this literature. It is not clear why the recommended approach of separating the dataset into the traditional testing and validation cohorts was not undertaken. I do recognise that the classifications lead to outcome profiles that have some "fidelity" with the expected outcomes of incident and prevalent MI, and therefore provides some assurance, but this still remains a single sample problem, and does not meet there standards of using the term validation.

Please see and apply principles from the TRIPOD recommendations

The use of coded data, rather than primary data sources (and the lack of validation) is problematic for the real clinical use of this algorithm. This should be recognised. This use of this algorithm is therefore useably for only cohort level or population level activities, rather than clinical (patient level) functions.

The inclusion of patient reported disease states is of uncertain value in this report. Some rationale for their inclusion should be provided.

6. PLOS authors have the option to publish the peer review history of their article (what does this mean?). If published, this will include your full peer review and any attached files.

Reviewer #1: No

---

## [Author Response · Author response to Decision Letter 0]

11 Feb 2022

Thank you for reviewing our manuscript.

Please see attached rebuttal letter for responses.

---

## [Editor Report · Decision Letter 1]

18 Feb 2022

Reproducible disease phenotyping at scale: example of coronary artery disease in UK Biobank

PONE-D-21-39043R1

Dear Dr. Denaxas,

We’re pleased to inform you that your manuscript has been judged scientifically suitable for publication and will be formally accepted for publication once it meets all outstanding technical requirements.

Kind regards,

Dylan A Mordaunt, MB ChB, MPH, MHLM, FRACP, FAIDH

Academic Editor

PLOS ONE

Additional Editor Comments (optional):

Thank you for your resubmission. This now meets the criteria for publication.
---

## [Editor Report · Acceptance letter]

24 Mar 2022

PONE-D-21-39043R1 

Reproducible disease phenotyping at scale: example of coronary artery disease in UK Biobank 

Dear Dr. Denaxas:

I'm pleased to inform you that your manuscript has been deemed suitable for publication in PLOS ONE. Congratulations! Your manuscript is now with our production department. 

Kind regards, 

on behalf of

Dr. Dylan A Mordaunt 

Academic Editor

PLOS ONE